# Sexual behaviour and STI testing among Dutch swingers: A cross-sectional internet based survey performed in 2011 and 2018

Carolina J. G. Kampman[1,2]*, Jeannine L. A. Hautvast[2], Femke D. H. Koedijk[1], Marieke E. M. Bijen[1], Christian J. P. A. Hoebe[3,4]

1 Public Health Service Twente, Enschede, The Netherlands, 2 Department of Primary and Community Care, Radboud University Medical Centre, Radboud Institute for Health Sciences, Nijmegen, The Netherlands, 3 Public Health Service South Limburg, Heerlen, The Netherlands, 4 Faculty of Health, Medicine and Life Sciences, Department of Social Medicine and Medical Microbiology, Care and Public Health Research Institute Maastricht, Maastricht University Medical Center (MUMC+), The Netherlands

* k.kampman@ggdtwente.nl

## Abstract

### Background

Swingers, heterosexuals who, as couples, practice mate swapping or group sex with other couples or heterosexual singles, are at risk for sexually transmitted infections (STIs). Therefore, the aim of this study was to assess changes in sexual behaviour and STI testing behaviour, as well as predictors of STI testing.

### Methods

Two cross-sectional studies were performed, using the same internet survey in 2011 and 2018. For trend analysis, sexual behaviour and STI testing behaviour were used. Socio-demographics, swinger characteristics, sexual behaviour, and psycho-social variables were used to assess predictors of STI testing in the past year, using multivariable regression analysis.

### Results

A total of 1173 participants completed the survey in 2011, and 1005 in 2018. Condom use decreased for vaginal (73% vs. 60%), oral (5% vs. 2%), and anal sex (85% vs. 75%). STI positivity was reported in 23% and 30% of the participants, respectively, although testing for STI was comparable between both years (~65%).

The following predictors of STI testing were significant: being female (OR = 1.9, 95%CI: 1.2–2.9), having a high swinging frequency (>12 times a year, OR = 3.7, 95%CI: 1.9–7.3), swinging at home (OR = 1.6, 95%CI: 1.0–2.7), receiving a partner notification (OR = 1.7, 95%CI: 1.2–2.6), considering STI testing important (OR = 4.3, 95%CI: 2.2–8.5), experiencing no pressure from a partner to test (OR = 0.6, 95%CI: 0.3–0.9), partners test for STI regularly (OR = 10.0, 95%CI: 6.2–15.9), perceiving STI testing as an obligation (OR = 2.1, 95%CI: 1.3–3.5), experiencing no barriers such as being afraid of testing (OR = 1.9, 95%CI: 1.2–

**Funding:** The author(s) received no specific funding for this work.

**Competing interests:** The authors have declared that no competing interests exist.

3.1), limited opening hours (OR = 1.6, 95%CI: 1.0–2.4), and forgetting to plan appointments (OR = 3.0, 95%CI: 2.0–4.6).

## Conclusions

Swingers exhibit self-selection for STI testing based on their sexual behaviour. However, STI prevention efforts are still important considering the increasing numbers of reported STIs, the decreased use of condom use, and the one-third of swingers who were not tested in the previous year.

## Introduction

Swingers are heterosexuals who, as couples, practice mate swapping or group sex with other couples or heterosexual singles. Although swingers self-identify as heterosexual, they frequently engage in same-sex sexual activities. Swingers are at risk for sexually transmitted infections (STIs), as they engage in unprotected sex with multiple sexual partners and substance misuse [1–3]. Swingers can transmit STIs within their own sexual network and to other sex partners outside their network through overlapping sexual partnerships. These concurrent sexual partnerships and potential bridging make them a target population of public health importance [1,4].

Only a few studies have estimated STI positivity rates among swingers. A Dutch study by Dukers et al found a *Chlamydia trachomatis* (CT) positivity rate of 8% and a *Neisseria gonorrhoea* (NG) positivity rate of 4% among swingers, which was lower than the STI positivity rate among all heterosexuals attending the STI clinic [5]. In the Dutch surveillance data, the overall STI positivity rate among swinger men was 16%, and 11% among swinger women [6]. A Belgian study by Platteau et al found that 81 out of 313 swingers who reported ever being tested for STI had ever had an STI diagnosis [7].

A Dutch study by Dukers et al showed that swingers take an STI test more often than men who have sex with men (MSM), or heterosexual men and women [5]. Another Dutch study by Spauwen et al showed that, overall, 72%, 62%, and 56% of swingers who consulted the STI clinic, reported that regular STI testing, partner notification, and condom use when engaging in sex, respectively, is the norm in the swinger community [8].

Before 2015, swingers were eligible for free and anonymous consultations at Dutch STI clinics. However, since 2015, based on the relatively low STI incidence, they were no longer eligible at STI clinics and have therefore been advised to consult a general practitioner (GP) for STI testing. This change in health policy since 2015 might hamper proper STI control in swingers, because STI testing at GPs is not free and anonymous, and swingers might refrain from identifying themselves as a swinger.

Lack of testing in swingers might implicate a potential rise in STI prevalence, and therefore testing behaviour among swingers is relevant as this might have a public health impact. To our knowledge, no studies have been conducted on to determine whether STI testing behaviour in swingers changes over time. Therefore, we performed cross-sectional studies in 2011 and 2018, using an internet survey, to compare sexual behaviour and STI testing behaviour, and to assess the influence of possible socio-demographic, behavioural, and psycho-social predictors of testing behaviour. The study outcomes can be used to evaluate current STI testing policy for swingers and provide information about the optimal STI clinic accessing policy and optimal STI test advice.

## Methods

### Study design, population, and data collection

Two cross-sectional studies were performed using an internet survey with the same questions in 2011 and in 2018. The content of the internet survey was developed based on information gathered in semi-structured interviews with swingers. The psycho-social variables were developed based on these interviews combined with the theory of planned behaviour [9–11]. The survey consisted of questions on socio-demography, swinger characteristics, sexual behaviour, STI test behaviour, and psycho-social determinants.

To recruit a broad sample of swingers in the Netherlands, both internet surveys were advertised at national websites that are frequently visited by swingers, including swinger websites, swinger club websites, and swinger dating websites. A banner with a link to the survey was published on the participating websites. Participants were requested to fill in the survey alone (not together as a couple) and were asked to only participate in the survey once per study. Participants who did not meet the definition of swinging (being part of a heterosexual couple and having sex with others, or being single and having sex with heterosexual couples), participants who were younger than 18 years, and those who did not swing in the past year were excluded from the analysis.

The incentive to participate in the study was the chance to win one of five dinner cheques with a value of 50 euros at the end of both study periods. Both internet surveys remained online for two months.

The survey software program Survey Monkey was used to embed the questions and provide the data for the analysis. Surveys that were not fully completed were excluded from analysis.

### Variables

Data on the following socio-demographic variables were collected: age at time of filling in survey, highest reported level of education (low educational level is pre-primary education; primary education or first stage of basic education; intermediate educational level is lower secondary education or second stage of basic education and high educational level is upper secondary education or tertiary education), gender, sexual preference, and relationship status (single or in a relationship). We combined the variables gender and sexual preference, as we expected sexual preference in men to be of greater public health importance than sexual preference in women.

Furthermore, the following swinger characteristics were analysed: swinging years (how many years engaged in swinging), swinging frequency (swinging how many times in the past year), and swinging location (at home, sexclub, hotel, party or holiday, answered by 'yes' or 'no').

The following sexual behaviour variables were collected: mean number of partners during swinging, ever received a partner notification for an STI during swinging period, having had condomless sex during vaginal, oral, and/or anal sex and when changing partners, ever had an STI during swinging period (chlamydia, gonorrhoea, syphilis, HIV, hepatitis B, genital warts, *Herpes genitalis*, *Trichomonas vaginalis*, and scabies were considered STIs), and drug and alcohol use during swinging.

Additionally, the following STI testing behaviour variables were collected: STI testing in the past year, STI testing location, and reasons for STI testing.

Lastly, psycho-social variables were collected as part of the following domains: STI risk perception, attitudes towards STI testing, social norm regarding STI testing, and self-efficacy and barriers regarding STI testing.

## Data analysis

We included only fully completed surveys in our data analysis. Descriptive analyses were performed for all variables, separately for both years. The $\chi^2$ test was used for testing differences in proportions between outcomes from 2011 and 2018. A p-value of <0.01 was considered to be statistically significant.

Univariable and multivariable logistic regression analyses were performed to identify predictors for the outcome measure 'STI testing in the past year', separately for both years. The results of the univariable and multivariable regression analysis were comparable between both years, except for the following predictors from the univariable regression analysis: gender, number of partners while swinging, condom change when changing partners, drug use, and the STI risk perception predictors 'Swinging partners don't have many STIs' and 'STI consequences are not severe'. Since most variables in the regression analyses for both years separately were comparable, a combined logistic regression analysis was performed to identify predictors for the outcome measure 'STI testing in the past year' for 2011 and 2018 together. As the demographic variables age and education were significantly different between 2011 and 2018, all predictors were adjusted for these demographics, as well as study year, in the combined logistic regression analyses. Backward logistic regression was used in multivariable analysis to further analyse the influence of predictors on STI testing. All variables with a p-value < 0.01 in univariable analyses were included. Predictors with p<0.01 were considered statistically significant in the multivariable analysis. Odds Ratios (ORs) and 99% Confidence Intervals (CIs) were presented to show the associations between the predictors and the outcomes in Table 2.

Analyses were conducted using SPSS for Windows, version 25.0 (IBM Inc., Somers, New York, United States).

## Medical ethical approval

The study was formally exempted from full medical ethical approval, as stated by the medical ethical committee of the Radboudumc Nijmegen (nr: 2018–4217) and according to Dutch Law. Data were obtained using the online survey tool 'Survey Monkey' and were registered in a fully anonymized and de-identified manner. To enter the prize pool for random allotment of dinner cheques, respondents were directed to a separate survey where they could enter their email address (only used for sending the incentive when applicable).

# Results

## Study population

In 2011, a total of 2152 participants started the survey, of which 1173 completed it (54.5%). In 2018, a total of 1478 participants started the survey, of which 1005 completed it (68.0%). Between both surveys, there were slight differences in the participating study population of swingers. In 2018, participating swingers were slightly older (mean age 43.4 years in 2011 vs. 46.5 years in 2018), had a higher educational level (59% vs. 50%), had slightly higher numbers of swinging years (mean 6.5 vs. 7.9 years), and had small differences in swinging locations (e.g. in 2011 84% were swinging at home vs. 79% in 2011). Gender, sexual preference, swinging frequency, relationship status, number of swinging partners and drug and alcohol use while swinging were equally distributed in both years; see Table 1.

## STI and sexual behaviour

Swingers who participated in 2018 reported having had an STI more often than swingers who participated in 2011 (23% vs. 30%). Furthermore, in 2018, participating swingers reported

**Table 1. Socio-demographic, sexual behaviour, STI testing behaviour and psycho-social variables of swingers in The Netherlands (2011, 2018).**

| | 2011 (n = 1173) | 2018 (n = 1005) | Total (n = 2178) | p value |
|---|---|---|---|---|
| | N(%) | N(%) | N(%) | |
| **Socio-demographic variables** | | | | |
| Age* | | | | **<0.001** |
| 18–30 | 109 (9.3) | 74 (7.4) | 183 (8.4) | |
| 31–40 | 290 (24.8) | 208 (20.7) | 498 (22.9) | |
| 41–50 | 542 (46.3) | 347 (34.6) | 889 (40.9) | |
| 51–60 | 203 (17.3) | 308 (30.7) | 511 (23.5) | |
| ≥61 | 27 (2.3) | 66 (6.6) | 93 (4.3) | |
| Education | | | | **<0.001** |
| Low educational level | 135 (11.5) | 66 (6.6) | 201 (9.3) | |
| Intermediate educational level | 452 (38.7) | 347 (34.7) | 799 (36.8) | |
| High educational level | 582 (49.8) | 588 (58.7) | 1170 (53.9) | |
| Gender and sexual preference (men) | | | | 0.019 |
| Bisexual men | 324 (27.6) | 311 (30.9) | 635 (29.9) | |
| Heterosexual men | 443 (37.8) | 402 (40.0) | 845 (38.8) | |
| Women | 406 (34.6) | 292 (29.1) | 698 (32.0) | |
| Relationship status | | | | 0.013 |
| Relationship | 1036 (88.3) | 851 (84.7) | 1887 (86.6) | |
| Single | 137 (11.7) | 154 (15.3) | 291 (13.4) | |
| **Swinger characteristics** | | | | |
| Swinging years* | | | | **<0.001** |
| 0–5 years | 658 (56.3) | 488 (48.6) | 1146 (52.7) | |
| 6–10 years | 326 (27.9) | 282 (28.1) | 608 (28.0) | |
| 11–20 years | 160 (13.7) | 187 (18.6) | 347 (16.0) | |
| ≥21 years | 25 (2.2) | 47 (4.7) | 72 (3.3) | |
| Swinging frequency | | | | 0.030 |
| 1–2 times a year | 161 (13.8) | 119 (11.9) | 280 (12.9) | |
| 3–12 times a year | 692 (59.0) | 635 (63.1) | 1327 (61.0) | |
| >12 times a year | 320 (27.2) | 251 (25.0) | 571 (26.2) | |
| Swinging location# | | | | |
| At home | 927 (79.0) | 847 (84.3) | 1774 (81.5) | **0.002** |
| Sexclub | 728 (62.1) | 561 (55.8) | 1289 (59.2) | **0.003** |
| Hotel | 194 (16.5) | 283 (28.1) | 477 (21.9) | **<0.001** |
| Party | 158 (13.5) | 144 (14.3) | 302 (13.9) | 0.563 |
| Holidays | 147 (12.5) | 115 (11.4) | 262 (12.0) | 0.436 |
| **Sexual behaviour variables** | | | | |
| No. partners during swinging^ | | | | 0.766 |
| 1–2 | 900 (76.7) | 643 (64.0) | 1543 (70.8) | |
| 3 or more | 252 (21.5) | 186 (18.5) | 438 (20.1) | |
| Ever received partner notification for an STI during swing period | | | | 0.289 |
| Yes | 508 (43.3) | 458 (45.6) | 966 (44.4) | |
| No | 665 (56.7) | 547 (54.4) | 1212 (55.6) | |
| Vaginal sex with condom during swinging | | | | **<0.001** |
| Always | 813 (72.7) | 589 (58.9) | 1402 (66.7) | |
| Not always | 306 (27.3) | 393 (40.0) | 699 (33.3) | |
| Oral sex with condom during swinging | | | | **<0.001** |
| Always | 55 (4.9) | 18 (1.8) | 73 (3.5) | |

*(Continued)*

**Table 1.** (Continued)

| | 2011 (n = 1173) | 2018 (n = 1005) | Total (n = 2178) | p value |
|---|---|---|---|---|
| | N(%) | N(%) | N(%) | |
| Not always | 1072 (95.1) | 964 (98.2) | 2036 (96.5) | |
| **Anal sex with condom during swinging** | | | | <0.001 |
| Always | 630 (84.6) | 499 (74.6) | 1129 (79.8) | |
| Not always | 115 (15.4) | 170 (25.4) | 258 (20.2) | |
| **Condom change when changing partners** | | | | **0.005** |
| Always | 987 (91.9) | 796 (88.2) | 1783 (90.2)) | |
| Not always | 87 (8.1) | 107 (11.8) | 194 (9.8) | |
| **Drug use during swinging*** | | | | 0.258 |
| Yes | 572 (48.8) | 513 (51.2) | 1085 (49.9) | |
| No | 601 (51.2) | 489 (48.8) | 1090 (50.1) | |
| **Alcohol use during swinging** | | | | 0.364 |
| Yes | 911 (77.7) | 764 (76.0) | 1675 (76.9) | |
| No | 262 (22.3) | 241 (24.0) | 503 (23.1) | |
| **Ever had an STI during swing period** | | | | <0.001 |
| Yes | 266 (22.7) | 298 (29.7) | 564 (25.9) | |
| No | 907 (77.3) | 707 (70.3) | 1614 (74.1) | |
| **STI testing variables** | | | | |
| **STI testing past year** | | | | 0.291 |
| Yes | 777 (66.2) | 644 (64.1) | 1421 (65.2) | |
| No | 396 (33.8) | 361 (35.9) | 757 (34.8) | |
| **STI testing location** | | | | <0.001 |
| STI clinic | 496 (63.8) | 291 (45.3) | 787 (55.5) | |
| General practitioner | 196 (25.2) | 260 (40.5) | 456 (32.1) | |
| Hospital | 76 (9.8) | 33 (5.1) | 109 (7.7) | |
| Home-test | 3 (0.4) | 47 (7.3) | 50 (3.3) | |
| Multiple test locations | 4 (0.5) | 6 (0.9) | 10 (0.7) | |
| Other | 2 (0.3) | 5 (0.8) | 7 (0.5) | |
| **Reasons for STI testing** | | | | 0.064 |
| Routine screening | 610 (78.5) | 476 (73.8) | 1086 (76.4) | |
| Partner notification | 47 (6.0) | 60 (9.3) | 107 (7.5) | |
| Unprotected sex | 46 (5.9) | 52 (8.1) | 98 (6.9) | |
| STI related symptoms | 31 (4.0) | 27 (4.2) | 58 (4.1) | |
| Other | 43 (5.5) | 30 (4.7) | 73 (5.1) | |
| **Psycho-social variables** | | | | |
| **STI risk perception** (%agree)$ | | | | |
| Risk of getting an STI is really small | 438 (53.9) | 327 (32.5) | 765 (35.1) | 0.019 |
| Swing partners don't have many STI | 632 (37.7) | 509 (50.6) | 1141 (52.4) | 0.132 |
| Swingers are a risk group for STI | 896 (76.4) | 814 (81.0) | 1710 (78.5) | **0.009** |
| STI consequences are not severe | 68 (5.8) | 42 (4.2) | 110 (5.1) | 0.086 |
| **Attitudes towards STI testing** (%agree)$ | | | | |
| STI testing is important for me | 999 (85.2) | 854 (85.0) | 1853 (85.1) | 0.901 |
| STI tests are unpleasant | 251 (21.4) | 222 (22.1) | 473 (21.7) | 0.696 |
| Testing as prevention | 71 (6.1) | 86 (8.6) | 157 (7.2) | <**0.024** |
| **Social norm regarding STI testing** (%agree)$ | | | | |
| Partners consider testing important | 1005 (85.7) | 879 (87.5) | 1884 (86.5) | 0.224 |
| Peer pressure to test | 704 (60.0) | 590 (58.7) | 1294 (59.4) | 0.535 |

*(Continued)*

**Table 1.** (Continued)

| | 2011 (n = 1173) | 2018 (n = 1005) | Total (n = 2178) | p value |
|---|---|---|---|---|
| | N(%) | N(%) | N(%) | |
| Partner pressure to test | 724 (61.7) | 606 (60.3) | 1330 (61.1) | 0.497 |
| Most swingers test for STI | 736 (62.7) | 561 (55.8) | 1297 (59.6) | **0.001** |
| Partner tests for STI regularly | 739 (63.0) | 592 (58.9) | 1331 (61.1) | 0.051 |
| STI testing is an obligation | 890 (75.9) | 768 (76.4) | 1658 (76.1) | 0.767 |
| **Self-efficacy and barriers regarding STI testing** (%agree)$ | | | | |
| Make time | 285 (24.3) | 217 (21.6) | 502 (23.0) | 0.135 |
| Afraid of needles | 132 (11.3) | 99 (9.9) | 431 (10.6) | 0.289 |
| Afraid of test result | 89 (7.6) | 62 (6.2) | 151 (6.9) | 0.194 |
| Afraid of test procedure | 92 (7.8) | 88 (8.8) | 180 (8.3) | 0.440 |
| Coming out as a swinger | 165 (14.1) | 209 (20.8) | 374 (17.2) | **<0.001** |
| Expensive | 148 (12.6) | 451 (44.9) | 599 (27.5) | **<0.001** |
| Afraid to see acquaintances | 126 (10.7) | 126 (12.5) | 252 (11.6) | 0.192 |
| Limited opening hours for STI testing | 272 (23.2) | 260 (25.9) | 532 (24.4) | 0.146 |
| Secrecy for steady partner | 46 (3.9) | 40 (4.0) | 86 (3.9) | 0.944 |
| Forget to make an appointment | 148 (12.6) | 151 (15.0) | 299 (13.7) | 0.104 |

percentages may not precisely add up to 100% due to rounding.

* Missings are not displayed.

# category 'other' swinging location was filled in by 42 participants in 2011 and 44 participants in 2018.

^ in 2018 169 participants had sex only with others and 30 participants had sex with own partner only in 2011 and 2018.

$ for these variables the indicated options were tested separately with a agree/disagree/neutral categorization, the selected % is shown in the title of the variable.

In bold: A p-value of <0.01 was considered to be statistically significant.

using a condom less often than participating swingers in 2011 (for example, 73% used a condom during vaginal sex vs. 59%); see Table 1.

## Predictors of STI testing in the past year

The predictors of the outcome measure 'STI testing in the past year' are shown in Table 2. In multivariable analysis, women tested more often for STIs in the past year than men (OR = 1.9, 95% CI 1.2 to 2.9). Furthermore, swingers who had a higher swinging frequency tested more often than swingers with a lower swinging frequency (OR = 3.7, 95% CI 1.9 to 7.3). Swingers who swing at home tested more often than swingers who do not swing at home (OR = 1.6, 95% CI 1.0 to 2.7). Furthermore, swingers who were notified of an STI by a partner during the swinging period tested more often than swingers who had not received a notification by a partner for an STI during the swinging period (OR = 1.7, 95% CI 1.2 to 2.6).

Concerning psycho-social variables related to STI testing, variables of the domains of attitude, social norm, and self-efficacy and barriers were significant predictors, whereas no variables of the risk perception domain were significant predictors. Important significant variables were that swingers who perceive STI testing to be important (OR = 4.3, 95% CI 2.2 to 8.5), who indicate that their partner tests for STIs regularly (OR = 10.0, 95% CI 6.2 to 15.9), and who perceive STI testing to be an obligation (OR = 2.1, 95% CI 1.3–3.5), tested more often for STIs than swingers who perceived differently. Otherwise, swingers who felt partner pressure to test had tested less often for STIs in the past year (OR = 0.6, 95% CI 0.3 to 0.9) than swingers who did not.

**Table 2. Predictors of STI testing in the past year among swingers in The Netherlands (2011 and 2018, n = 2178), adjusted for year, age and education.**

| | Univariable analysis aOR(99% CI) | Multivariable analysis aOR(99% CI) |
|---|---|---|
| **Socio-demographic variables** | | |
| Gender and sexual preference (men) | | |
| Bisexual men | 1.3 (1.0–1.7) | 1.2 (0.8–1.9) |
| Heterosexual men | ref **1.7 (1.2–2.2)** | ref **1.9 (1.2–2.9)** |
| Women | | |
| Relationship status | | |
| Relationship | ref | nt |
| Single | 1.1 (0.8–1.6) | |
| **Swinger characteristics** | | |
| Swinging years | | |
| 0–5 years | ref | nt |
| 6–10 years | 1.1 (0.8–1.4) | |
| 11–20 years | 1.1 (0.7–1.5) | |
| ≥21 years | 1.0 (0.5–2.0) | |
| Swinging frequency | | |
| 1–2 times a year | ref | ref |
| 3–12 times a year | **2.9 (2.0–4.1)** | **2.0 (1.1–3.5)** |
| >12 times a year | **6.3 (4.1–9.6)** | **3.7 (1.9–7.3)** |
| Swinging location (ref = no)* | | |
| At home | **2.8 (2.1–3.7)** | **1.6 (1.0–2.7)** |
| Sexclub | 0.8 (0.6–1.0) | nt |
| Hotel | 1.3 (1.0–1.7) | nt |
| Party | **1.6 (1.1–2.3)** | ns |
| Holidays | 1.3 (0.9–1.9) | nt |
| **Sexual behaviour variables** | | |
| No. partners during swinging | | |
| 1–2 | ref | ns |
| 3 or more | **1.5 (1.1–2.1)** | |
| Ever received partner notification during swing period | | |
| Yes | **3.8 (2.9–4.9)** | **1.7 (1.2–2.6)** |
| No | ref | ref |
| Vaginal sex with condom during swinging | | |
| Always | ref | nt |
| Not always | 1.0 (0.8–1.4) | |
| Oral sex with condom during swinging | | |
| Always | ref | nt |
| Not always | 1.2 (0.6–2.3) | |
| Anal sex with condom during swinging | | |
| Always | ref | nt |
| Not always | 0.9 (0.6–1.3) | |
| Condom change when changing partners | | |
| Always | ref | nt |
| Not always | **0.6 (0.4–0.9)** | |
| Drug use during swinging | | |
| Yes | **2.9 (2.2–3.7)** | 1.3 (0.9–2.0) |

(*Continued*)

**Table 2.** (Continued)

| | Univariable analysis aOR(99% CI) | Multivariable analysis aOR(99% CI) |
|---|---|---|
| No | ref | ref |
| Alcohol use during swinging | | |
| Yes | **0.7 (0.5–0.9)** | nt |
| No | ref | |
| Ever had an STI during swinging-period | | |
| Yes | **3.5 (2.5–4.8)** | 1.4 (0.9–2.3) |
| No | ref | ref |
| **Psycho-social variables** | | |
| **STI risk perception** (ref = agree/neutral)* | | |
| Risk of getting an STI is small | **1.9 (1.5–2.5)** | ns |
| Swing partners don't have many STI | 1.2 (0.8–1.9) | nt |
| Swingers are a risk group for STI | 0.8 (0.5–1.2) | nt |
| STI consequences are not severe | **1.4 (1.0–1.8)** | ns |
| **Attitudes towards STI testing** (ref = disagree/neutral)* | | |
| STI testing is important for me | **15.4 (10.1–23.5)** | **4.3 (2.2–8.5)** |
| STI tests are unpleasant | **0.6 (0.4–0.8)** | ns |
| Testing as prevention | **0.6 (0.4–1.0)** | ns |
| **Social norm regarding STI testing** (ref = disagree/neutral)* | | |
| Partners consider testing important | **10.8 (7.2–16.2)** | ns |
| Peer pressure to test | **2.9 (2.3–3.7)** | ns |
| Partner pressure to test | **5.2 (4.0–6.6)** | **0.6 (0.3–0.9)** |
| Most swingers test for STI | **4.0 (3.1–5.1)** | ns |
| Partner tests for STI regularly | **16.5 (12.3–22.1)** | **10.0 (6.2–15.9)** |
| STI testing is an obligation | **8.1 (6.0–10.9)** | **2.1 (1.3–3.5)** |
| **Self-efficacy and barriers regarding STI testing** (ref = agree/neutral)* | | |
| Make time | **3.6 (2.8–4.6)** | ns |
| Afraid of needles | **2.8 (2.1–3.7)** | ns |
| Afraid of test result | **2.1 (1.6–2.8)** | ns |
| Afraid of test procedure | **5.4 (4.1–7.2)** | **1.9 (1.2–3.1)** |
| Coming out as swinger | **3.8 (2.9–4.9)** | ns |
| Testing is expensive | **2.4 (1.8–3.1)** | ns |
| Afraid to see acquaintances | **2.6 (1.9–3.3)** | ns |
| Limited opening hours | **2.8 (2.2–3.5)** | **1.6 (1.0–2.4)** |
| Secrecy steady partner | **3.5 (2.5–5.0)** | ns |
| Forget to make appointment | **5.4 (4.1–7.0)** | **3.0 (2.0–4.6)** |

ref, reference.

*for these variables the indicated options were each tested separately with a yes/no or agree/disagree/neutral categorization, the reference being shown in the title of the variable.

In bold: Significant (p<0.01 in univariable and p<0.01 in multivariable analysis).

Furthermore, swingers who indicated not being afraid of the test procedure (OR = 1.9, 95% 1.2 to 3.1), did not perceive limited opening hours for STI testing (OR = 1.6, 95%CI 1.0 to 2.4), and who indicated not forgetting to make an appointment (OR = 3.0, 95%CI 2.0 to 4.6) tested more often for STIs than swingers who indicated having opposite opinions regarding these issues.

## Discussion

### Statement of principal findings

Our study of two cross-sectional Dutch surveys showed that swingers reported reduced use of condoms in 2018 (for example, in 2011, 73% used a condom during vaginal sex, compared to 59% in 2018) and reported having had an STI more often (23% versus 30%) than swingers who participated in 2011. However, a similar majority of swingers reported testing for STIs in both years (66% in 2011 and 64% in 2018) and regarded testing for STIs as important (85% in 2011 and 85.0% in 2018). We thus recognize an increased STI positivity rate and increase in sexual risk behaviour between 2011 and 2018 in swingers, although testing behaviour remained the same.

The following predictors for STI testing in swingers were assessed and appeared to be positive: swingers with a higher swinging frequency (>12 times a year OR = 3.7), swingers who were notified of an STI by a partner (OR = 1.7), swingers who swing at home (OR = 1.6), swingers who feel that STI testing is an obligation (OR = 2.1), swingers whose partners test for STIs regularly (OR = 10.0), and swingers who state that STI testing is important (OR = 4.3) tested for STI more often.

### Strengths and weaknesses

A strength of this study was that a large number of swingers participated in the survey in both years (55% and 68%). Due to changed public health policy, it has become more difficult for swingers to be tested at STI clinics after 2015, and swingers are therefore harder to reach for studies through STI clinics. Posting the advertisement online has proven to be effective in reaching swingers and has shown the willingness among swingers to participate in research and voice their opinions.

Another strength of this study is the measurement of psycho-social variables, such as STI risk perception, attitudes, social norms, and self-efficacy regarding STI testing besides the more often measured sexual behaviour variables. With the use of these variables, clearer insight has been obtained into reasons and beliefs of swingers possibly influencing STI testing behaviours. Addressing these reasons in public health messages might lower existing barriers for swingers who are still hesitant to undergo testing, even though almost two-thirds are already regularly tested.

However, a general limitation of our study is a possible sampling bias. First, only swingers who visit a swinger dating website were invited to participate. As a consequence, generalizability to the entire population of swingers in the Netherlands might be affected, although we know from field work and other studies that most swingers are registered at these websites.

Though we did perform semi-structured interviews with swingers and used the theory of planned behaviour as input for our survey, we did not validate our survey. Therefore, we do not know for sure if our survey is measuring what we intended to measure. Our results should be read bearing this in mind.

Third, STI diagnosis was self-reported over their period of swinging years, though self reported STI history may not be an appropriate proxy for true STI history. Therefore, self-reported STI diagnosis mighthamper translation into the prevalence or incidence of STI [12].

Lastly, in this study, no identifying information was available, and therefore we do not know if the same swingers participated in both surveys. Study findings show that participants in 2018 were older and reported more swinging years than those in 2011. This might indicate that some swingers participated in both years, which might have led to overestimation of some outcomes.

## Comparison to other studies

This study shows that the majority of participating swingers tested for STIs in the past year (66.2% in 2011 and 64.1% in 2018). A Canadian study, however, stated that swingers 'rarely' access STI health services (< 40.8% visited STI health services) [13]. Since public health policy changed in 2015, it is more difficult for swingers to access STI clinics in the Netherlands. This change in policy is reflected in our study findings, which shows that instead of testing at an STI clinic, swingers report visiting their general practitioner or ordering a home test more often than in 2011. This is in line with national data on declining STI clinic attendance of swingers [6].

Furthermore, this study shows a substantial percentage of self-reported STI diagnosis during swinging years (22.7% in 2011 and 29.7% in 2018). This finding is in line with a Belgian study performed by Platteau et al, which reported that 26% of the swingers have had an STI [7]. Several other studies have reported about STI positivity rates among swingers, but they reported lower STI incidences ranging from 8 to 13%, because of a shorter time span in which the STI was diagnosed or reported [1,5,7]. Our study shows that swingers who have a higher swinging frequency and those who were notified by a partner during a swinging period, tested more often for STI. There are no other studies that have found a similar relationship. There are, however, studies that have shown associations between a high STI positivity rate among swingers who have received a partner notification, swingers who had STI related symptoms, swingers with a previous STI, and swingers who had unprotected sex [5,8,13].

Our study also assessed psycho-social variables as predictors for STI testing in the past year. There are no other studies on psycho-social predictors of STI testing among swingers. There are, however, studies among students on predictors of STI testing. These studies show that attitude was positively associated with STI testing among students, as were perceived social norms towards STI testing, high STI risk perception, and the absence of perceived STI test barriers. These findings are in line with our study, except for high STI risk perception, as, in our multi-variable analysis, we did not have significant results in this domain [14,15].

Our study also shows that within the domain of self-efficacy and barriers, swingers who are not afraid of the test procedure, who do not experience limited opening hours, and who do not forget to make an appointment test for STIs, tested more often than those swingers experiencing the opposite. There are no studies that have assessed self-efficacy and barriers towards STI testing among swingers, but there are studies performed on self-efficacy and test barriers among MSM. One Canadian study among MSM reported that perceived lack of health knowledge among testing providers and limited clinical capacity were two major barriers towards STI testing [16]. A Dutch study among MSM found that burdensome testing procedures, among others, was a barrier towards STI testing [17].

## Significance of the study

Our study findings show that swingers in the Netherlands test for STIs regularly, even after the change in public health policy that made swingers no longer eligible at STI clinics for free and anonymous STI testing. Therefore, reconsidering this changed public health policy on swingers has proven to be of lesser need.

Although swingers test for STIs regularly, the location of STI testing has changed, when comparing study results of 2011 and 2018. Participating swingers in 2018 reported visiting a GP more than participating swingers in 2011. However, as studies show that GPs may omit testing for all STIs and all body locations, especially when swingers do not identify themselves as such and being MSM while swinging, which means that education is needed for GPs [18–23].

Participating swingers in 2018 reported making use of a home-based STI test more often than participating swingers in 2011. Home-based testing has advantages, such as a wider reach, being anonymous, and no need to travel for an STI test. However, there are also downsides to home-based testing, such as poor quality of the STI test and lack of opportunity to obtain sexual health counselling [24–26]. Fortunately, a list of test facilities proven to be of good quality is already present in the Netherlands. Monitoring these online and home-based test facilities will continue to be needed in the future.

It is of concern that swingers did report a higher STI positivity rate during their swinging years, when comparing results from 2011 with 2018. However, participants in 2018 were older and had more swinging years than participants in 2011. Therefore, participating swingers in 2018 had a greater time period to report having had an STI than participating swingers in 2011. However, condom use with any kind of sex had decreased when comparing 2011 to 2018. These findings indicate that primary prevention targeting swingers to prevent them from getting STIs is still needed.

## Conclusion

This study shows that two-third of swingers tested for STIs in the past year. STI testing is perceived as important, and barriers for testing such as fear or logistical challenges are infrequently reported. Swingers show a self-selection for STI testing based on their sexual risk behaviour, such as swingers who receive a partner notification and swingers with a high swinging frequency undergoing more testing for STIs. Taking swingers into account as a target group for STI prevention efforts is still important considering the high reported STI positivity rate, the decreased use of condoms, and the one-third of swingers who were not tested in the previous year.

## Supporting information

**S1 File. Survey 2011.**
(PDF)

**S2 File. Survey 2018.**
(PDF)

**S3 File. Swingers data 20112018 minimal deidentified dataset.**
(SAV)

**S4 File. Swingers data 20112018 minimal deidentified dataset.**
(XLSX)

## Acknowledgments

We would like to acknowledge the participating swinger websites during our 2011 and 2018 survey. Furthermore, we would like to thank Editage (www.editage.com) for English language editing.

## Author Contributions

**Conceptualization:** Jeannine L. A. Hautvast, Marieke E. M. Bijen, Christian J. P. A. Hoebe.

**Data curation:** Carolina J. G. Kampman, Femke D. H. Koedijk, Marieke E. M. Bijen.

**Formal analysis:** Femke D. H. Koedijk, Marieke E. M. Bijen.

**Investigation:** Carolina J. G. Kampman, Christian J. P. A. Hoebe.

**Methodology:** Carolina J. G. Kampman, Jeannine L. A. Hautvast, Femke D. H. Koedijk, Christian J. P. A. Hoebe.

**Supervision:** Jeannine L. A. Hautvast, Femke D. H. Koedijk, Christian J. P. A. Hoebe.

**Visualization:** Marieke E. M. Bijen, Christian J. P. A. Hoebe.

**Writing – original draft:** Carolina J. G. Kampman.

**Writing – review & editing:** Carolina J. G. Kampman, Jeannine L. A. Hautvast, Femke D. H. Koedijk, Christian J. P. A. Hoebe.

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
