## [Decision Letter · Decision Letter 0]

27 May 2020

PONE-D-20-05060

Trends and predictors of STI testing behaviour among Dutch swingers; a cross-sectional internet based survey performed in 2011 and 2018

PLOS ONE

Dear Dr. Kampman,

Thank you for submitting your manuscript to PLOS ONE. After careful consideration, we feel that it has merit but does not fully meet PLOS ONE’s publication criteria as it currently stands. Therefore, we invite you to submit a revised version of the manuscript that addresses the points raised during the review process.

We look forward to receiving your revised manuscript.

Kind regards,

Henry F. Raymond

Academic Editor

PLOS ONE

Additional Editor Comments (if provided):

Thank you for your submission. Although we strive to have papers reviewed by two or more reviewers the current times have limited availability. Based on the one review we received and my reading of your paper I recommend undertaking a major revision in line with the reviewers comments.

2. Please address the following:

- Please refrain from stating p values as 0.00, either report the exact value or employ the format p<0.001.

- Please refer to any post-hoc corrections to correct for multiple comparisons during your statistical analyses. If these were not performed please justify the reasons. Please refer to our statistical reporting guidelines for assistance (https://journals.plos.org/plosone/s/submission-guidelines.#loc-statistical-reporting).

- Please include additional information regarding the survey or questionnaire used in the study and ensure that you have provided sufficient details that others could replicate the analyses. For instance, if you developed a questionnaire as part of this study and it is not under a copyright more restrictive than CC-BY, please include a copy, in both the original language and English, as Supporting Information.

Reviewers' comments:

Reviewer's Responses to Questions

**Comments to the Author**

1. Is the manuscript technically sound, and do the data support the conclusions?

Reviewer #1: Partly

2. Has the statistical analysis been performed appropriately and rigorously? 

Reviewer #1: Yes

3. Have the authors made all data underlying the findings in their manuscript fully available?

Reviewer #1: No

4. Is the manuscript presented in an intelligible fashion and written in standard English?

Reviewer #1: Yes

5. Review Comments to the Author

Reviewer #1: This paper takes on an interesting topic – sexually transmitted infection testing among swingers – and provides online survey results from two time points in the Netherlands. There are a few areas where clarifying language and expanding analyses can improve the paper, as described below. Additionally, more information is needed about how the data can be accessed beyond just instructions to contact the corresponding author, unless more detail is added to the data availability statement.

Major:

The title and introduction reference trends, but a formal trend analysis (e.g. Cochran-Armitrage test for trend) does not seem to have been included. Adding one to the existing chi-square tests (or clarifying if the described chi-square tests are tests for trend) should be a priority in the revision.

Provide confidence intervals for estimates.

Please clarify whether the multivariable model includes year. If not, please justify, as it seems that year would be an important factor to adjust for if interested in changes over time.

How were the questions for the survey written and chosen? More details about development of the survey would be helpful. Were any of the questions validated? If not, this needs to be mentioned in the limitations.

Why is gender and sexual orientation combined into one variable? It would make sense to consider these separately, and the rationale for combining them is not clear.

To the degree it is possible, it would be helpful to assess for missingness not at random. Did the people who started but did not complete the survey systematically differ from those who did? If so, predictors of missingness need to be included in the multivariable model (e.g., if younger people were less likely to complete the survey, then having age in the multivariable model will decrease bias in estimator).

Pay attention to terminology. I have highlighted a few examples, but it would be worthwhile to review the whole paper to ensure use of consistent, updated terminology. E.g., Line 5: Rephrase sentence to avoid stigmatizing term “risky sexual behavior” (and tautology of risky behavior leading to risk) and instead focus on describing what the behaviors are and stating whether the literature suggests swingers engage in the behaviors more often than non-swingers. Additionally, replace the term “substance abuse” with the more precise/less stigmatizing “substance misuse” (behavior) or “substance use disorder” (diagnosis) as appropriate. Review the rest of the paper for this kind of terminology and update.

Was “protected” defined for participants? Does it include only condoms or also contraception (or in 2018, HIV pre-exposure prophylaxis)? It would be more precise to use the term “condomless” than “unprotected” but this should be balanced with preserving the language from the survey itself.

Similarly, how was a “swing period” defined?

It would be helpful to provide the full text of the survey itself as an appendix to answer these and previous questions noted.

The limitations section should potentially note a few other limitations based on the wording of questions described above. It would be worthwhile to cite a reference about discrepancies between reported vs. laboratory confirmed STI results there.

Would remove “rational” from describing decision making – it is non-specific and implies a value judgment – calls to question, what would “irrational self-selection” be? A better term could be something like risk-based, or risk-informed.

Minor:

Throughout, use verb “tested” instead of “performed testing” to refer to the action taken by people. As in: Swingers surveyed in 2018 tested for STIs more frequently than swingers surveyed seven years earlier.

In the intro, since same studies are referenced a few times, it would be helpful to identify them by first author’s last name.

Line 112: State what the higher education level is.

Line 114: Clarify what is meant by proportion at home (e.g., “Significantly more swingers reported swinging at home in 2018 (84%) compared to 2011 (79%).”

Table 1: Clarify “partner notification” refers to STI diagnosis and if applicable, specify if this includes partner-partner notification as well as third-party notification (e.g., via health dept). If this is in the questionnaire, providing the questionnaire is sufficient.

6. PLOS authors have the option to publish the peer review history of their article (what does this mean?). If published, this will include your full peer review and any attached files.

Reviewer #1: No

---

## [Author Response · Author response to Decision Letter 0]

8 Jul 2020

Editor comments 

Thank you for your submission. Although we strive to have papers reviewed by two or more reviewers the current times have limited availability. Based on the one review we received and my reading of your paper I recommend undertaking a major revision in line with the reviewers comments.

Thank you for reading our paper and giving us a chance to improve our manuscript. We will address each point below. 

We have ensured that our manuscript meets PLOS ONE’s requirements.

2. Please address the following:

a. Please refrain from stating p values as 0.00, either report the exact value or employ the format p<0.001.

We have replaced p values 0.00 with p<0.001 throughout our manuscript. We have reported all our p-values in the same manner.

b. Please refer to any post-hoc corrections to correct for multiple comparisons during your statistical analyses. If these were not performed please justify the reasons. Please refer to our statistical reporting guidelines for assistance (https://journals.plos.org/plosone/s/submission-guidelines.#loc-statistical-reporting).

We did not perform any post-hoc correction, due to the reason that we already lowered our p value to <0.01 instead of <0.05 to correct for multiple comparisons.

c. Please include additional information regarding the survey or questionnaire used in the study and ensure that you have provided sufficient details that others could replicate the analyses. For instance, if you developed a questionnaire as part of this study and it is not under a copyright more restrictive than CC-BY, please include a copy, in both the original language and English, as Supporting Information.

We will include a copy of the questionnaire in the original language as ‘Supporting Information’. Unfortunately, we do not have an English translation. Therefore, we elaborated on the topics in our ‘methods’ section by describing variables more extensively, on page 4, as follows:

“Data on the following socio-demographic variables were collected: age at time of filling in survey, highest reported level of education (low educational level is pre-primary education, primary education or first stage of basic education, intermediate educational level is lower secondary education or second stage of basic education and high educational level is upper secondary education or tertiary education), gender, sexual preference, and relationship status (single or in a relationship). We combined the variables gender and sexual preference, because of an already large list of variables and because sexual preference in men is possibly of greater public health importance than sexual preference in women.

Furthermore, the following swinger characteristics were analysed: swinging years (how many years engaged in swinging), swinging frequency (swinging how many times in the past year), and swinging location (at home, sexclub, hotel, party or holiday, answered by ‘yes’ or ‘no’).

The following sexual behaviour variables were collected: mean number of partners during swinging, ever received a partner notification for an STI during swinging period, having had condomless sex during vaginal, oral, and/or anal sex and when changing partners, ever had an STI during swinging period (chlamydia, gonorrhoea, syphilis, HIV, hepatitis B, genital warts, Herpes genitalis, Trichomonas vaginalis, and scabies were considered STIs), and drug and alcohol use during swinging. 

Additionally, the following STI testing behaviour variables were collected: STI testing in the past year, STI testing location, and reasons for STI testing. 

Lastly, psycho-social variables were collected as part of the following domains: STI risk perception, attitudes towards STI testing, social norm regarding STI testing, and self-efficacy and barriers regarding STI testing.”

d. We note that you have indicated that data from this study are available upon request. PLOS only allows data to be available upon request if there are no legal or ethical restrictions on sharing data publicly. For information on unacceptable data access restrictions, please see http://journals.plos.org/plosone/s/data-availability#loc-unacceptable-data-access-restrictions

There are no restrictions on sharing a minimal de-identified data set. We will upload our a minimal de-identified data set as Supporting Information. 

We reason that publication of the minimal de-identified dataset is a negligible risk to participant confidentiality, because only 2 demographic variables were included in our dataset; age group and gender and sexual preference. Other demographic variables of the participants were not included in the minimal de-identified dataset.

We have added the following paragraph in our ‘data availability statement’ section on page 16;

“Data availability statement

Consent for publication of raw data is not obtained, but the dataset is fully anonymous in a manner that can easily be verified by any user of the dataset. Publication of the dataset clearly and obviously presents minimal risk to confidentiality of study participants.”

We have also addressed this altered data availability statement in our revised cover letter.

The ‘data not shown’ phrase concerns the subheading under table 1, page 7. What we meant is that this data is not shown in our table, but we do show the data in the subheading. Therefore, ‘data not shown’ is redundant and deleted from our manuscript.

Reviewer comments

1. Is the manuscript technically sound, and do the data support the conclusions?

Reviewer #1: Partly

2. Has the statistical analysis been performed appropriately and rigorously? 

Reviewer #1: Yes __

3. Have the authors made all data underlying the findings in their manuscript fully available?

Reviewer #1: No

4. Is the manuscript presented in an intelligible fashion and written in standard English?

Reviewer #1: Yes

5. Review Comments to the Author

Reviewer #1: This paper takes on an interesting topic – sexually transmitted infection testing among swingers – and provides online survey results from two time points in the Netherlands. There are a few areas where clarifying language and expanding analyses can improve the paper, as described below. Additionally, more information is needed about how the data can be accessed beyond just instructions to contact the corresponding author, unless more detail is added to the data availability statement.

Thank you very much for carefully reading our manuscript and for providing us with your useful comments. We will address your comments point by point below.

Major:

a. The title and introduction reference trends, but a formal trend analysis (e.g. Cochran-Armitrage test for trend) does not seem to have been included. Adding one to the existing chi-square tests (or clarifying if the described chi-square tests are tests for trend) should be a priority in the revision.

In our data analysis we used Pearson chi-square tests to test for differences in proportions between demographic outcomes from 2011 and 2018. We did not use the chi-square test for trends, because there were only two measure points (year 2011 and year 2018). This does however (which we realised in hindsight after reading your comments) not fully qualify as a ‘trend’ and does not justify using the chi-square for trends. Therefore, we adjusted our title on page 1, as follows:

“STI testing and sexual behaviour among Dutch swingers; a cross-sectional internet based survey performed in 2011 and 2018”

Also, we have improved our ‘introduction’ section on page 3 as follows:

“Lack of testing in swingers might implicate a potential rise in STI prevalence, and therefore testing behaviour among swingers is relevant as this might have a public health impact. To our knowledge, no studies have been conducted on to determine whether STI testing behaviour in swingers changes over time. Therefore, we performed cross-sectional studies in 2011 and 2018, using an internet survey, to compare sexual behaviour and STI testing behaviour, and to assess the influence of possible socio-demographic, behavioural, and psycho-social predictors of testing behaviour. The study outcomes can be used to evaluate current STI testing policy for swingers and provide information about the optimal STI clinic accessing policy and optimal STI test advice.”

b. Provide confidence intervals for estimates.

We have already provided confidence intervals for the estimates in our ‘results’ section on page 9 and in table 2 on page 10. We have added confidence intervals in our ‘results’ section of our abstract on page 2. 

c. Please clarify whether the multivariable model includes year. If not, please justify, as it seems that year would be an important factor to adjust for if interested in changes over time.

We did adjust our multivariable model for year (2011/2018), as stated in the heading in table 2, page 9:

“Table 2. Predictors of STI testing in the past year among swingers in The Netherlands (2011 and 2018, n=2178), adjusted for year, age and education”

d. How were the questions for the survey written and chosen? More details about development of the survey would be helpful. Were any of the questions validated? If not, this needs to be mentioned in the limitations.

We elaborated about the development in our ‘methods’ section on page 4 as follows:

“The content of the internet survey was developed based on information gathered in semi-structured interviews with swingers. The psycho-social variables were developed based on these interviews combined with the theory of planned behaviour [9,10,11]. The survey consisted of questions on socio-demography, swinger characteristics, sexual behaviour, STI test behaviour, and psycho-social determinants. ”

The questions were not validated, we have added a limitation in our ‘discussion’ section on page 12 as follows:

“Though we did perform semi-structured interviews with swingers and used the theory of planned behaviour as input for our survey, we did not validate our survey. Therefore, we do not know for sure if our survey is measuring what we intend to measure. Our results should be read bearing this in mind.”

e. Why is gender and sexual orientation combined into one variable? It would make sense to consider these separately, and the rationale for combining them is not clear.

It is a common in the Netherlands to pair these two variables, we have added a section on this in our ‘methods’ section on page 4 as follows: 

“We combined the variables gender and sexual preference, as we expected sexual preference in men to be of greater public health importance than sexual preference in women.”

f. To the degree it is possible, it would be helpful to assess for missingness not at random. Did the people who started but did not complete the survey systematically differ from those who did? If so, predictors of missingness need to be included in the multivariable model (e.g., if younger people were less likely to complete the survey, then having age in the multivariable model will decrease bias in estimator).

We only included fully completed surveys in our data analysis. We have added this sentence in our ‘methods’ section on page 5 as follows to clarify this:

“We included only fully completed surveys in our data analysis.”

The missings that are present in table 1 are participants who did not fill in a question, because the question was not applicable to them.

g. Pay attention to terminology. I have highlighted a few examples, but it would be worthwhile to review the whole paper to ensure use of consistent, updated terminology. E.g., Line 5: Rephrase sentence to avoid stigmatizing term “risky sexual behavior” (and tautology of risky behavior leading to risk) and instead focus on describing what the behaviors are and stating whether the literature suggests swingers engage in the behaviors more often than non-swingers. Additionally, replace the term “substance abuse” with the more precise/less stigmatizing “substance misuse” (behavior) or “substance use disorder” (diagnosis) as appropriate. Review the rest of the paper for this kind of terminology and update.

We have reviewed our paper and we have replaced stigmatizing sentences with less stigmatizing sentences in our manuscript, including your examples, as follows:

Page 3: “Swingers are at risk for sexually transmitted infections (STIs), as they engage in unprotected sex with multiple sexual partners and substance misuse.”

Page 15: “Swingers show a self-selection for STI testing based on sexual risk behaviour , such as swingers who receive a partner notification and swingers with a high swinging frequency undergoing more testing for STIs.”

h. Was “protected” defined for participants? Does it include only condoms or also contraception (or in 2018, HIV pre-exposure prophylaxis)? It would be more precise to use the term “condomless” than “unprotected” but this should be balanced with preserving the language from the survey itself.

In our survey the participants were asked if they used a condom with various sexual behaviours. We have replaced ‘protected’ with ‘condom use’ throughout our manuscript as well as in our tables.

i. Similarly, how was a “swing period” defined?

Swinging period was defined as how many years one engaged in swinging. We have added this to our ‘methods’ section on page as follows:

“Furthermore, the following swinger characteristics were analysed: swinging years (how many years engaged in swinging), swinging frequency (swinging how many times in the past year), and swinging location.”

j. It would be helpful to provide the full text of the survey itself as an appendix to answer these and previous questions noted.

We will upload a copy of our survey as a supplementary file. As our survey is only available in Dutch language, we have elaborated more on the variables used in our ‘methods’ section on page 5, as follows:

“Data on the following socio-demographic variables were collected: age at time of filling in survey, highest reported level of education (low educational level is pre-primary education, primary education or first stage of basic education; intermediate educational level is lower secondary education or second stage of basic education; and high educational level is upper secondary education or tertiary education), gender, sexual preference, and relationship status (single or in a relationship). We combined the variables gender and sexual preference, because of an already large list of variables and because sexual preference in men is possibly of greater public health importance than sexual preference in women.

Furthermore, the following swinger characteristics were analysed: swinging years (how many years engaged in swinging), swinging frequency (swinging how many times in the past year), and swinging location (at home, sexclub, hotel, party or holiday, answered by ‘yes’ or ‘no’).

The following sexual behaviour variables were collected: mean number of partners during swinging, ever received a partner notification for an STI during swinging period, having had condomless sex during vaginal, oral, and/or anal sex and when changing partners, ever had an STI during swinging period (chlamydia, gonorrhoea, syphilis, HIV, hepatitis B, genital warts, Herpes genitalis, Trichomonas vaginalis, and scabies were considered STIs), and drug and alcohol use during swinging.”

k. The limitations section should potentially note a few other limitations based on the wording of questions described above. It would be worthwhile to cite a reference about discrepancies between reported vs. laboratory confirmed STI results there.

We have added 2 limitations in our ‘discussion’ section on page 12 based on previous questions of the reviewer (see text below). And we have provided the limitation addressing self-reported versus laboratory confirmed STI results with a reference, as follows:

“ Though we did perform semi-structured interviews with swingers and used the theory of planned behaviour as input for our survey, we did not validate our survey. Therefore, we do not know if our survey is measuring what we were meaning to measure. Our results should be read bearing this in mind.

Third, STI diagnosis was self-reported over their period of swinging years, though self-reported STI history may not be an appropriate proxy for true STI history. Therefore, self-reported STI diagnosis might hamper translation into the prevalence or incidence of STI [12].”

l. Would remove “rational” from describing decision making – it is non-specific and implies a value judgment – calls to question, what would “irrational self-selection” be? A better term could be something like risk-based, or risk-informed.

We have deleted the word ‘rational’ in our abstract on page 2 and in our conclusion on page 15. By deleting the word ‘rational’ we remove a value whether or not it is a good self-selection. Our conclusion on page 15 is as follows:

“Swingers show a self-selection for STI testing based on their sexual risk behaviour, such as swingers who receive a partner notification and swingers with a high swinging frequency undergoing more testing for STIs.”

Minor:

a. Throughout, use verb “tested” instead of “performed testing” to refer to the action taken by people. As in: Swingers surveyed in 2018 tested for STIs more frequently than swingers surveyed seven years earlier.

We have removed the word “performed testing’ with “tested” throughout our manuscript.

b. In the intro, since same studies are referenced a few times, it would be helpful to identify them by first author’s last name.

We identified the authors of Dukers (ref. 5), Platteau (ref. 7) and Spauwen (ref. 8) in our introduction on page 3 and in our discussion on page 13.

c. Line 112: State what the higher education level is.

We have explained the definition of the education levels in our ‘methods’ section on page 4, as follows:

“Data on the following socio-demographic variables were collected: age at time of filling in survey, highest reported level of education (low educational level is pre-primary education, primary education or first stage of basic education, intermediate educational level is lower secondary education or second stage of basic education and high educational level is upper secondary education or tertiary education), gender, sexual preference, and relationship status (single or in a relationship).”

d. Line 114: Clarify what is meant by proportion at home (e.g., “Significantly more swingers reported swinging at home in 2018 (84%) compared to 2011 (79%).”

We have elaborated on swinging location in our ‘methods’ section on page 5, as follows:

“Furthermore, the following swinger characteristics were analysed: swinging years (how many years engaged in swinging), swinging frequency (swinging how many times in the past year), and swinging location (at home, sexclub, hotel, party or holiday, answered by ‘yes’ or ‘no’).”

Also, we have explained the example we gave on swinging location in our ‘results’ section on page 6 better, as follows:

“In 2018, participating swingers were slightly older (mean age 43.4 years in 2011 vs. 46.5 years in 2018), had a higher educational level (59% vs. 50%), were more often single (15% vs. 12%), had slightly higher numbers of swinging years (mean 6.5 vs. 7.9 years), and had small differences in swinging locations (e.g. in 2011 84% were swinging at home vs. 79% in 2011).”

e. Table 1: Clarify “partner notification” refers to STI diagnosis and if applicable, specify if this includes partner-partner notification as well as third-party notification (e.g., via health dept). If this is in the questionnaire, providing the questionnaire is sufficient.

In table 1 on page 7 we have referred to receiving a partner notification for STI, as follows:

“Ever received partner notification for an STI during swing period”

Furthermore, we elaborated on the partner notification variable in our ‘methods’ section on page 5, as follows:

“ever received a partner notification for an STI during swinging period”

Unfortunately we did not specify the partner notification in partner-partner notification and third party notifications.

6. PLOS authors have the option to publish the peer review history of their article (what does this mean?). If published, this will include your full peer review and any attached files.

Do you want your identity to be public for this peer review? For information about this choice, including consent withdrawal, please see our Privacy Policy.

Reviewer #1: No

---

## [Editor Report · Decision Letter 1]

14 Sep 2020

STI testing and sexual behaviour among Dutch swingers; a cross-sectional internet based survey performed in 2011 and 2018

PONE-D-20-05060R1

Dear Dr. Kampman,

We’re pleased to inform you that your manuscript has been judged scientifically suitable for publication and will be formally accepted for publication once it meets all outstanding technical requirements.

Kind regards,

Henry F. Raymond

Academic Editor

PLOS ONE
---

## [Editor Report · Acceptance letter]

22 Sep 2020

PONE-D-20-05060R1 

Sexual behaviour and STI testing among Dutch swingers; a cross-sectional internet based survey performed in 2011 and 2018 

Dear Dr. Kampman:

I'm pleased to inform you that your manuscript has been deemed suitable for publication in PLOS ONE. Congratulations! Your manuscript is now with our production department. 

Kind regards, 

on behalf of

Dr. Henry F. Raymond 

Academic Editor

PLOS ONE